# Noisy Information Bottlenecks for Generalization

## Abstract

We propose Noisy Information Bottlenecks (NIB) to limit mutual information between learned parameters and the data through noise. We show why this benefits generalization and allows mitigation of model overfitting both for supervised and unsupervised learning, even for arbitrarily complex architectures. We reinterpret methods including the Variational Autoencoder, $\beta$-VAE, network weight uncertainty and a variant of dropout combined with weight decay as special cases of our approach, explaining and quantifying regularizing properties and vulnerabilities within information theory.

## 1 Introduction

Bayesian inference is at the core of many directions of machine learning research in recent years. Applications range from a principled treatment of uncertainty in neural networks (Gal, 2016) over latent variable models for high-dimensional data (Kingma & Welling, 2013; Rezende & Mohamed, 2015) to reinterpretations of popular stochastic regularizers (Kingma et al., 2015; Gal, 2016) that inspired more flexible extensions (Louizos & Welling, 2017; Molchanov et al., 2017). In all but the simplest models, exact inference is intractable, so approximations are necessary. Among those, variational inference (Wainwright et al., 2008) has been particularly popular, as it is amenable to gradient-based stochastic optimization (Kingma & Welling, 2013; Titsias & Lázaro-Gredilla, 2014; Ranganath et al., 2014) and thus highly scalable.

Typically, researchers seek to develop more flexible approximate posterior distributions (Rezende & Mohamed, 2015; Kingma et al., 2016; Salimans et al., 2015; Ranganath et al., 2016; Huszár, 2017; Chen et al., 2018; Vertes & Sahani, 2018; Burda et al., 2015; Cremer et al., 2017) in the hope of more faithfully representing the true posterior. However, recent works have suggested (Trippe & Turner, 2018; Braithwaite & Kleijn, 2018; Shu et al., 2018) that restricting the family of variational approximations can, in fact, have a positive regularizing effect, leading to better generalization.

In this work, we seek to provide an information theoretic explanation for the observed behaviour of different families variational approximations. By reinterpreting Gaussian mean-field inference as maximum a-posteriori in a noisy model, we can quantify bounds on the mutual information between the data and the parameters variables. We explore the potential of this approach for regularization. Unlike methods relying on information-based objectives (Tishby et al., 2000; Shamir et al., 2010; Agakov & Barber, 2004; Chen et al., 2016; Phuong et al., 2018), it is compatible with standard variational inference.

## 2 Noisy Information Bottlenecks

This section characterizes model overfitting within information theory and points out a mitigation strategy that limits the amount of information extracted by the inferred model about the data. We will show this to be implicitly deployed by a common class of inference techniques in section 3 in order to explain observed regularizing effects.

### 2.1 An Information-Theoretic Characterization of Model Overfitting

Intuitively, model overfitting can be understood as memorization or learning too much information about the training data. More formally, if $D$ denote the random variable for all our data and $\theta$ all

learned parameters and latent variables in a given probabilistic model, we can quantify the amount of information gained about the data $D$ given a particular parameter value $\theta^*$ in

$$I(D, \theta = \theta^*) := H(D) - H(D|\theta = \theta^*) \tag{1}$$

where $H(D|\theta = \theta^*) := -\mathbb{E}_{p(D|\theta = \theta^*)} \log p(D|\theta = \theta^*)$ denotes the entropy of a variable $D$ conditioned on a particular value $\theta^*$.

In the context of variational inference, model overfitting can now be characterized by the symptom that the learned information about the data $D$

$$\mathbb{E}_{\theta^* \sim q(\theta)} I(D, \theta = \theta^*) \tag{2}$$

is too large in expectation under the approximately inferred posterior distribution $q(\theta) \approx p(\theta|D)$.

We now motivate this objective through the extreme case of maximum-likelihood learning in the limit of unrestrictedly expressive models for unsupervised tasks. Here, $q$ is a point-mass on the parameter values that maximize the likelihood of the data. Under the assumption of indepenent and identically distributed (iid) datapoints, the model will then store all information that can possibly be extracted from the data, as shown in Appendix A. This is all information except the datapoint identity (the information necessary to distinguish between distinct training samples, e. g. the index) which cannot be learned due to the iid assumption. This will result in perfect overfitting up to the sample identity, thereby impeding generalization.

In unsupervised learning, information that can be learned by the model about the data will not be learned by the latent, a phenomenon known as the information preference problem (Chen et al., 2016; Alemi et al., 2016; Zhao et al., 2017; Phuong et al., 2018). Again, this can be characterized by the quantity from Equation 2 being too high.

## 2.2 WHY BOUNDING MUTUAL INFORMATION?

This observation motivates placing a limit on the expected amount of extracted information given by Equation 2. We observe that the expectation of $I(D, \theta = \theta^*)$ under the prior $p(\theta)$ is just the mutual information $I(D, \theta)$ as commonly defined in literature:

$$\mathbb{E}_{\theta^* \sim p(\theta)} I(D, \theta = \theta^*) = H(D) - \mathbb{E}_{\theta^* \sim p(\theta)} H(D|\theta = \theta^*) = H(D) - H(D|\theta) = I(D, \theta) \tag{3}$$

This implies that if we could limit $I(D, \theta)$ to not be greater than some capacity $C$, we would obtain the following guarantee: If our dataset $d$ was sampled from the marginal $d \sim p(D) = \int d\theta p(\theta) p(D|\theta)$, then a sample from the exact posterior $\theta^* \sim p(\theta|D = d)$ will, on average, contain no more information about the data $D$ than our capacity $C$ due to

$$\mathbb{E}_{d \sim p(D)} \mathbb{E}_{\theta^* \sim p(\theta|D=d)} I(D, \theta = \theta^*) = \mathbb{E}_{\theta^* \sim p(\theta)} I(D, \theta = \theta^*) = I(D, \theta) \leq C \tag{4}$$

This would limit the quantity given in Equation 2 in expectation under the assumptions that the model captures the nature of the generating process in $p(D)$ and that our posterior estimate $q(\theta)$ is close to the true posterior $p(\theta|D)$. These are common assumptions necessary to justify any variational Bayesian approach.

We derive further motivation for our approach from the fact that limiting mutual information between data and learned parameters provably bounds generalization error (Xu & Raginsky, 2017; Bassily et al., 2018). While the approach of limiting the amount of extracted information is already used for the purpose of generalization in adaptive analysis (Feldman & Steinke, 2018; Smith, 2017; Russo & Zou, 2015), in the following section we give an approach compatible with a variety of variational and deep learning algorithms.

## 2.3 NOISY INFORMATION BOTTLENECKS

How can we limit $I(D, \theta)$? Due to nonlinearities in typical deep models, it is hard to calculate and therefore limit the mutual information between data and parameters directly. Instead, one way to achieve this is to make the data dependent only on some noisy version $\tilde{\theta}$ of the learned variables $\theta$, resulting in a model $p(\theta, \tilde{\theta}, D) = p(\theta) p(\tilde{\theta}|\theta) p(D|\tilde{\theta})$ forming a Markov chain as shown in Figure 1a.

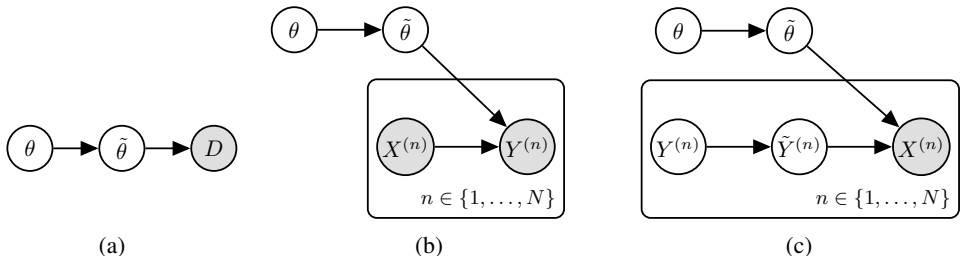

Figure 1: A noisy information bottleneck (a) limits the mutual information $I(D, \theta)$ between the data $D$ and the learned variables $\theta$ by making the data only depend on a corrupted version $\tilde{\theta}$ of the latent, with a limited $I(\tilde{\theta}, \theta)$. It can be applied to both (b) supervised with inputs $X^{(n)}$ and labels $Y^{(n)}$ and (c) generative models with latents $Y^{(n)}$ and datapoints $X^{(n)}$.

The prior $p(\theta)$ and corruption process $p(\tilde{\theta}|\theta)$ are chosen to result in a certain finite $I(\tilde{\theta}, \theta)$. The data processing inequality (Cover & Thomas, 2012) then ensures

$$I(D, \theta) \leq I(\tilde{\theta}, \theta) \tag{5}$$

for any model architecture $p(D|\tilde{\theta})$. $C = I(\tilde{\theta}, \theta)$ now acts as a capacity, bounding the expected amount of information that is extracted about the data. In the next section, we will show that such a bound is implicit in an important class of variational inference techniques. As we will see, the corruption process will be realized through injected noise, and we therefore term this architecture *noisy information bottleneck* (NIB). Quantifying the corresponding capacity $C$ will help to explain observed regularizing effects in both supervised and unsupervised learning (as shown in Figure 1b and 1c) from an information-theoretic perspective.

## 3 REGULARIZATION THROUGH CONSTRAINED INFERENCE

Most variational approximation schemes such as variational autoencoders (Kingma et al., 2015; Rezende & Mohamed, 2015) and network weight uncertainty (Blundell et al., 2015) use simple Gaussian mean field inference for tractable training using the reparameterization trick. In this section, we quantify information-theoretic capacity constraints implicit in Gaussian mean field inference that helps to understand observed regularizing effects. We will first illustrate the bound for a fixed-scale distribution and then discuss the more general case where the variance is learned as well.

### 3.1 FIXED-SCALE GAUSSIAN MEAN FIELD INFERENCE

Let $p(\theta, D)$ denote a model with parameters $\theta$ and data $D$. The free energy objective now is

$$F(q) = \mathbb{E}_{q(\theta)} \log p(D|\theta) - D_{\mathrm{KL}}(q(\theta)||p(\theta)) \tag{6}$$

We consider the case of Gaussian mean field inference with fixed variance $\sigma^2$ on parameters with component-wise independent priors $p(\theta) = \mathcal{N}(0, \sigma_p^2 I)$. We can write the inferred $q(\theta)$ in its reparameterized form $\theta = \mu^* + \sigma\epsilon$, with $\mu^*$ being the inferred mean and $\epsilon$ being noise sampled from $\mathcal{N}(0, I)$, as shown in Figure 2a. We can then write the objective as

$$F(\mu^*) = \mathbb{E}_{p(\epsilon)} \log p(D|\theta = \mu^* + \sigma\epsilon) - \sum_i \frac{\mu_i^{*2}}{2\sigma_p^2} + \mathrm{const.} \tag{7}$$

which is optimized with respect to $\mu^*$. We use $i \in \{1, \ldots, K\}$ to denote the model parameter index.

To show how Gaussian mean field contains an implicit information bottleneck we include the inferred means $\mu$ into the generative model, without changing the resulting objective. We therefore define a model $p'(\mu, \tilde{\mu}, D) = p'(\mu)p'(\tilde{\mu}|\mu)p'(D|\tilde{\mu})$ (Figure 2b), where $\mu \sim \mathcal{N}(0, \sigma_p^2 I)$ represents the parameters of our new model and $\tilde{\mu} = \mu + \sigma\epsilon$ its noise-injected version with $\epsilon \sim \mathcal{N}(0, I)$. We leave the rest of the model $p'(D|\tilde{\mu}) = p(D|\theta)$ unchanged. This model is an instantiation of a noisy information bottleneck as in Figure 1a, where $\mu$ and $\tilde{\mu}$ take the roles of $\theta$ and $\tilde{\theta}$.

Figure 2: Gaussian mean field inference of fixed scale on model parameters with a Gaussian prior (a) can be reinterpreted as MAP inference on a model with injected noise (b): The mean of the original inference model correspond to the parameters of the new generative model.

On this new model, we do variational inference by sampling noise $\epsilon$ from the prior $q'(\epsilon) = p'(\epsilon)$, while learning a deterministic approximate posterior $q'(\mu)$ with point-mass at a particular $\mu^*$, resulting in a MAP-like objective[1]

$$F'(q') = \mathbb{E}_{q'(\mu)q'(\epsilon)} \log p'(D|\mu, \epsilon) + \sum_i \log p'(\mu_i^*) \tag{8}$$

Using above definitions, and Equation 7, this implies equivalence $F(\mu^*) = F'(\mu^*)$ of both objectives, resulting in exactly equivalent training algorithms.

For the new model, we get $H(\tilde{\mu}) = \frac{K}{2} \log 2\pi e \left(\sigma^2 + \sigma_p^2\right)$ and $H(\tilde{\mu}|\mu = \mu^*) = \frac{K}{2} \log 2\pi e \sigma^2$ for any $\mu^*$, inducing a noisy information bottleneck of capacity

$$I(\tilde{\mu}, \mu) = H(\tilde{\mu}) - \mathbb{E}_{\mu^* \sim p(\mu)} H(\tilde{\mu}|\mu = \mu^*) = \frac{K}{2} \log \left(1 + \frac{\sigma_p^2}{\sigma^2}\right) \tag{9}$$

in the sense of Equation 5, where $K$ denotes the number of parameters. This quantity is known as the capacity of channels with Gaussian noise in signal processing (Cover & Thomas, 2012). Intuitively, a high prior variance $\sigma_p^2$ corresponds to a large capacity, while a high noise variance $\sigma^2$ reduces it. Simply adjusting the signal-to-noise ratio $\frac{\sigma_p^2}{\sigma^2}$ allows to create an information bottleneck of any desired capacity. This suggests the usefulness of applying fixed-scale Gaussian mean field inference to model parameters for regularization.

There are relations to existing approaches: When applied to training neural networks, MAP training on this model can be interpreted as applying weight decay combined with a additive noise $\mathcal{N}(0, \sigma^2)$ on all network weights. Molchanov et al. (2017) shows that this results in multiplicative noise on the unit activations. Wang & Manning (2013) reports that empirical results do not change significantly when dependencies between the different elements of the layer output are ignored, which is then equivalent to scaled Gaussian dropout (Kingma et al., 2015).

## 3.2 FLEXIBLE-SCALE GAUSSIAN MEAN FIELD INFERENCE

The variance in Gaussian mean field inference is typically learned for each parameter (Kingma et al., 2015; Rezende & Mohamed, 2015; Blundell et al., 2015). We can obtain a capacity constraint for this case by regarding both the inferred mean $\mu$ and variance $\sigma^2$ as the new latent. For simplicity and different from the fixed-scale case, we assume a prior variance of $\sigma_p^2 = 1$.

The derivation is similar to the one for the fixed-scale case from the previous section and given in a slightly different form in Appendix C. The resulting capacity is not in closed analytic form, but we obtain a numeric result of $0.45$ bits per latent component. The derivation can be generalized to approaches where the KL term from the objective is scaled by some factor $\beta > 0$ as done by $\beta$-VAE (Higgins et al., 2017) in the context of amortized inference on latent variables. Curiously, this results

---

[1]Due to the inferred noise $q'(\epsilon)$, this is not pure MAP, but rather a lower bound objective similar to the free energy. We refer to it as MAP for compactness, and connections to the variational free energy are given in Appendix B.

in priors $\mu_i \sim \mathcal{N}\left(0, \frac{1}{\beta}\right)$ and $\sigma_i^2 \sim \Gamma\left(\frac{\beta}{2} + 1, \frac{\beta}{2}\right)$. We observe that higher $\beta$ corresponds to smaller capacity, which is given by the mutual information $I(\tilde{\mu}_i, (\mu_i, \sigma_i^2))$ between our new latent $(\mu_i, \sigma_i^2)$ and $\tilde{\mu}_i$. This formalizes the intuition that a higher weight of the complexity term in our objective increases regularization by limiting the capacity. Numeric results for the capacity are shown in Figure 8 in the appendix.

### 3.3 Should Variational Inference be Flexible or Regularizing?

As mentioned in the introduction, more flexible approximate inference is the goal of many recent approaches. However, in previous subsections we showed that training with Gaussian mean field inference can be reinterpreted as MAP learning on a noise-injected model that has a certain information-theoretic capacity that is useful to mitigate overfitting. Therefore the goals of good inference and regularization are in conflict.

The reinterpretation necessary to retrieve the capacity also suggests a natural resolution: Instead of using MAP on the noise-injected model, we can deploy arbitrarily flexible variational inference techniques. This separates out the concern of regularization into the model and allows to combine powerful inference with good regularization. The focus of this piece of work, however, is to demonstrate that limiting the mutual information between the model parameters and the data leads to better generalization, hence we leave exploring the use of powerful inference techniques in noisy models as future research.

## 4 Experiments

In VAEs, Gaussian mean field inference on the latents leads to a restricted latent capacity, but leaves the capacity of the model unbounded. This leaves VAEs vulnerable to model overfitting, as shown in subsection 2.1, and setting $\beta$ as done in (Higgins et al., 2017) is not sufficient to control complexity. Relatedly, the bound given through Equation 9 was also noted by Braithwaite & Kleijn (2018), but was again applied only to the latents and not the model parameters.

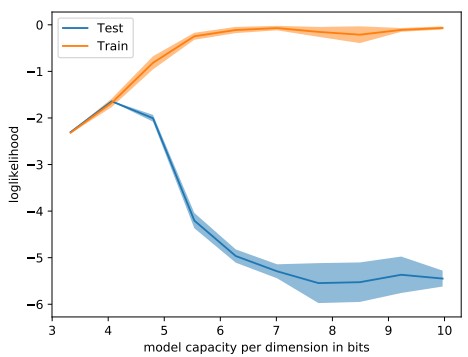

We therefore propose to apply Gaussian mean field inference of fixed scale to the model parameters in order to mitigate model overfitting both for supervised and unsupervised tasks. This is further motivated by the regularizing effect empirically observed when using Gaussian mean field inference over point estimates reported by Blundell et al. (2015). Fixing the variance $\sigma^2$ allows to easily control the capacity of the bottleneck, as described in subsection 3.1.

Figure 3: Classifying CIFAR10 with varying model capacities. Large capacities lead to overfitting while small capacities drown the signal in noise. Each configuration has been evaluated 5 times; mean and standard deviation are displayed.

In this section we validate this idea. In order to clearly show overfitting, we train large architectures and a small number of training samples for most experiments. We explore the effect noisy information bottlenecks implicit in fixed scale Gaussian mean field inference for varying model capacities, architectures and priors as well as training set sizes.

### 4.1 Supervised Learning

We apply our approach to classification on the CIFAR10 dataset, where we train on a subset of the first 5000 samples. We use 6 3x3 convolutional layers with 128 channels each followed by a relu activation function, every second of which implements striding 2 to reduce the input dimensionality.

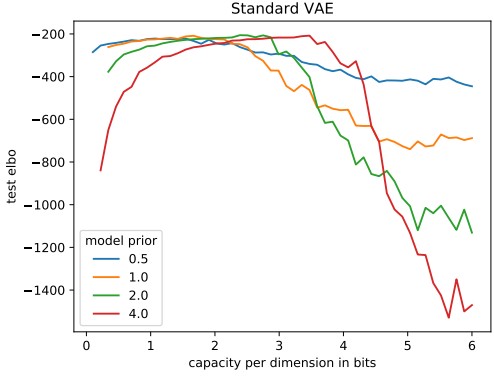
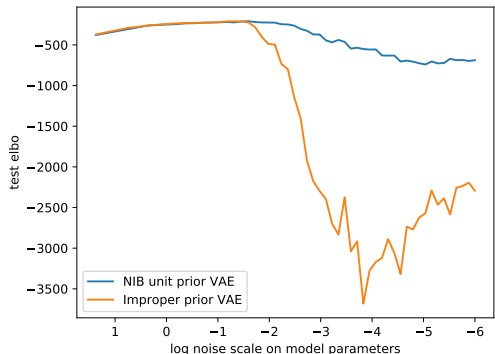

(a) The test ELBO is not invariant when varying the prior on the model parameters. Nevertheless, the first increasing and then decreasing trend when changing the capacity remains.

(b) Using an improper prior, similar to just using Gaussian Dropout on the weights, leads to accelerated decreasing generalization for smaller noise scales.

Figure 4: MNIST test reconstruction with a VAE training on 200 samples for various priors and capacities.

Finally, the last layer is a linear projection which parameterizes a categorical distribution. The capacity of each parameters in this network is set to specific values given by Equation 9.

Figure 3 shows that decreasing the model capacity per channel (by increasing the noise) reduces the training log-likelihood and increases the test loglikelihood until both of them meet at an optimal capacity. As noted before, without this prior the capacity would be infinite. We will analyze the phenomenon in the unsupervised case in the next subsection. It is also observable that very small capacities lead to a signal that is too noisy and good predictions are no longer possible. All these observations are in accordance with our predictions.

## 4.2 Unsupervised Learning

We now evaluate the regularizing effect of fixed-scale Gaussian mean field inference in an unsupervised setting. Therefore, we use a VAE (Kingma & Welling, 2013) with 2 latent dimensions and a 3-layer neural network parameterizing the conditional factorized Gaussian distribution. As usual, it is trained using the free energy objective. Again, we use a small training set of 200 examples for most experiments.

In our first experiment we analyze generalization by inspecting the test ELBO when varying the model capacity which can be seen in Figure 4a. Similar to the supervised case we can observe that there is a certain model capacity range that explains the data very well while less or more capacity results in noise drowning and overfitting respectively. In the same figure we also investigated whether the information-theoretic model capacity can predict generalization independently of the specific prior distribution. From Figure 4a we can conclude that while model capacity seems to influence generalization very strongly, it seems like model choice such as the prior on the parameters also affects generalization slightly. It also implies that weight decay (Krogh & Hertz, 1992) of fixed scale without parameters noise is not sufficient to regularize arbitrarily large networks. Nevertheless, the shapes of all ELBO-capacity dependencies are similar. In Figure 4b we investigated the extreme case of dropping the prior entirely and switching to ML learning instead by using an improper uniform prior. This approach is very similar to Gaussian dropout, as described in subsection 3.1. Dropping the prior sets the bottleneck capacity to infinity and should lead to worse generalization. Comparing the test ELBO to NIB in Figure 4b confirms this result for larger capacities. For larger noise scales, generalization is still working well, a result that is not explained in our information-theoretic framework, but plausible due to the deployed limited architecture.

Figure 5a shows how our approach affects the test ELBO for varying amounts of training data. Models with very small capacity extract less information from the data into the model, thus yielding a good test ELBO somewhat independent of the dataset size. This is visible as a graph that ascends

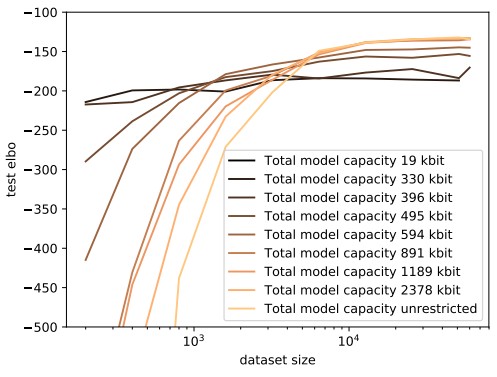 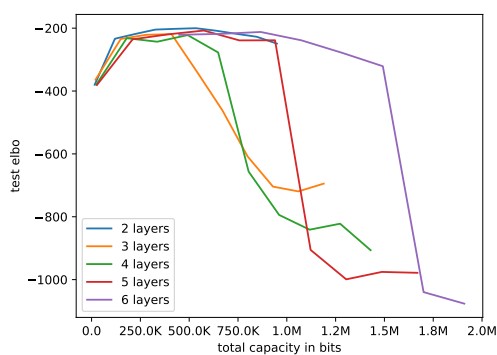

(a) Varying the number of samples. Depending on the size of the dataset higher capacities of the model are required to fit all the datapoints.

(b) Varying architecture. Overfitting is not getting worse for more layers if capacity is low enough. More layers do overfit only for higher capacities.

Figure 5: MNIST test reconstruction with a VAE training on varying dataset sizes, architectures, and model capacities.

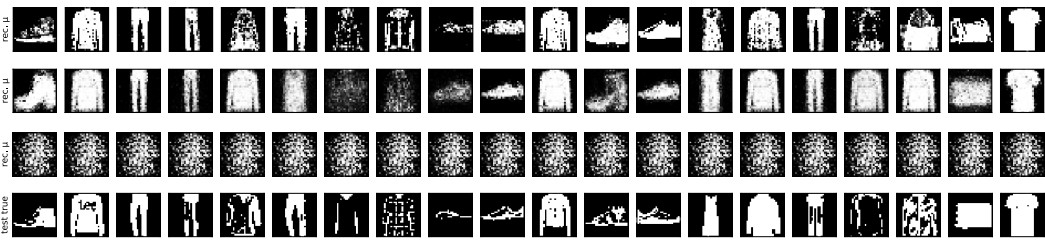

Figure 6: Test reconstruction means for binarized fashion MNIST trained on 200 samples with per-parameter capacities 5, 2 and 1 bits (from top) compared to the true data (bottom).

very little with more training data (e.g. total model capacity 330 kbits). Note that we here report the capacity of the entire model, which is the sum of the capacities for each parameter. In order to improve the test ELBO, more information from the data has to be extracted into the model. But clearly, this leads to non-generalizing information being extracted when the dataset is small, leading to overfitting, and generalizing information being extracted for larger datasets. This is visible as a strongly ascending test ELBO with larger dataset sizes and bad generalization for small datasets. We can therefore conclude that the information bottleneck needs to be chosen based on the amounts of data that is available. This is expected as we want to extract more information into the model the more information is available.

Furthermore, we inspected how the size of the model (here in terms of number of layers) affects generalization in Figure 5b. The generalization does not decrease for the same total capacity but larger networks deteriorate the performance for larger total capacities. This indicates that the total capacity is less important than the individual capacity (i.e. noise) per parameter. Nevertheless, larger networks are more prone to overfitting for very large model capacities. This makes sense as their functional form is less constrained, an aspect that is not captured by our framework.

Finally, we plot test reconstruction means for the binarized fashion MNIST dataset under the same setup for various capacities in Figure 6. In accordance with previous experiments, we observe that if the capacity is chosen too small, the model is not learning anything useful, while too large capacities result in overconfidence, which can be seen through most means being close to either 0 or 1. An intermediate capacity, on the other hand, makes sensible predictions (given that it was trained only on 200 samples) with sensible uncertainty, visible through gray pixels that correspond to high entropy.

## 5 RELATED WORK

We have shown that NIB is a practical approach to regularize supervised and unsupervised models, and that in contrast to existing approaches, it successfully regularizes models with a fixed parameter setting, largely independent of the depth of a network. Unlike existing regularization techniques, our approach features a capacity that can be naturally interpreted as a limit on the amount of information extracted about the given data by the inferred model.

The Information Bottleneck principle by Tishby et al. (2000); Shamir et al. (2010) aims to find a representation $Z$ of some input $X$ that is most useful to predict an output $Y$. For this purpose, the objective is to maximize the amount of information $I(Y, Z)$ the representation contains about the output under a bounded amount of information $I(X, Z)$ about the input:

$$\max_{I(X,Z)<C} I(Y, Z) \tag{10}$$

They describe a training procedure using the softly constrained objective

$$\min \mathcal{L}_{IB} = \min I(X, Z) - \beta I(Y, Z) \tag{11}$$

where $\beta > 0$ controls the trade-off.

Alemi et al. (2016) suggests a variational approximation for this objective. For the task of reconstruction, where labels $Y$ are identical to inputs $X$, this results exactly in the $\beta$-VAE objective (Achille & Soatto, 2017; Alemi et al., 2018). This is in accordance with our result from subsection 3.2 that there is a maximum capacity per latent dimension in the reinterpreted version of $\beta$-VAE that get smaller for greater $\beta$.

Reconstruction implicit in amortized inference for unsupervised learning is only one of many possible objectives for extracting representations, and it can be viewed a proxy for a not-yet-known future task. In this case, applying NIB to the latents is a way to enforce the hard constraint $I(X, Z) < C$ of objective Equation 10 directly through the structure of the model, removing the need to augment the objective with a soft constraint, as in Equation 11.

For the supervised case where the task is already known, as in Tishby et al. (2000), we could achieve a similar hard constraint by constructing a noisy information bottleneck layer anywhere in the network based on the channel capacity theorem (Cover & Thomas, 2012) by limiting input variance (e. g. through a tanh activation) and injecting noise as in NIB. This allows extracting representations useful for a given output task with standard variational training, while naturally constraining mutual information with the input.

The Information Bottleneck principle is concerned with the information contained in the latent representation. NIB limits mutual information with all inferred variables, namely latents and model parameters in the case of unsupervised learning. This paper focuses on characterizing and mitigating the vulnerability of existing learning algorithms due to unconstrained mutual information of the data with the model parameters.

## 6 CONCLUSION

We have quantified the regularizing effects observed in Gaussian mean field approaches from an information-theoretic perspective. We have explored the usefulness of the implicit noisy information bottleneck for the purpose of generalization. Our approach features a capacity that can be naturally interpreted as a limit on the amount of information extracted about the given data by the inferred model. We validated its practicality for both supervised and unsupervised learning. We have shown that the approach allows to improve generalization even for arbitrarily large networks.

While this work demonstrates the capability of NIB to mitigate model overfitting when applied to model parameters, inspecting the effect of a limited latent capacity is left for future work. We expect our approach to be compatible with powerful inference techniques while keeping up regularization guarantees. This is still to be confirmed experimentally.

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

## A    MAXIMUM-LIKELIHOOD ON UNRESTRICTED MODELS

We consider the extreme case of maximum-likelihood learning on $N$ identically distributed (iid) data in the limit of unrestrictedly expressive models. For simplicity, we assume discrete data to avoid an undefined empirical differential entropy. A similar argument could be made for continuous data. We can then write

$$I(D, \theta = \theta^*) = N I(X, \theta = \theta^*) \tag{12}$$

where $p(X|\theta)$ is the modeled distribution $p(X^{(n)}|\theta)$ of any data sample $X^{(n)}$. This definition is valid because the distribution is independent of the data sample index $n$ due to the iid assumption. Let $p(\hat{X})$ denote the empirical distribution. The maximum-likelihood objective can then be written as

$$\theta_{ML} = \arg\min_{\theta} D_{\mathrm{KL}}\left(p(\hat{X}) || p(X|\theta = \theta^*)\right) \tag{13}$$

In the unrestricted limit (e.g. arbitrarily sized network architecture), there exists a $\theta^*$ so that the distribution $p(X|\theta)$ is equal to $p(\hat{X})$. Because $D_{\mathrm{KL}}\left(p(\hat{X}) || p(X|\theta = \theta^*)\right)$ is then 0 and therefore minimal, this is actually $\theta_{ML}$. From $p(X|\theta = \theta_{ML}) = p(\hat{X})$ follows that the entropy conditioned on $\theta_{ML}$ becomes the empirical entropy

$$H(X|\theta = \theta_{ML}) = H(\hat{X}) \tag{14}$$

Thus,

$$I(X, \theta = \theta_{ML}) = H(X) - H(X|\theta = \theta_{ML}) = H(X) - H(\hat{X}) \tag{15}$$

This implies that all information about the data is learned by the model up to the identity of the training sample, which remains as uncertainty in the prediction.

## B    MAP AND MAXIMUM-LIKELIHOOD AS VARIATIONAL INFERENCE

Variational inference constrained to deterministic approximate posterior distributions with (continuous) probability mass function

$$q(Z) = \begin{cases} 1 & \text{for } Z = Z^* \\ 0 & \text{else} \end{cases} \tag{16}$$

(= point-mass at $Z^*$) can be interpreted as MAP:

$$\arg\min_{Z^*} F(q) \tag{17}$$

$$= \arg\min_{Z^*} D_{\mathrm{KL}}\left(q(Z) || p(Z|X)\right) \tag{18}$$

$$= \arg\min_{Z^*} \mathbb{E}_q\left(\log q(Z) - \log p(Z|X)\right) \tag{19}$$

$$= \arg\min_{Z^*} \log q(Z^*) - \log p(Z^*|X) \tag{20}$$

$$= \arg\min_{Z^*} - \log p(Z^*|X) \tag{21}$$

$$= \arg\max_{Z^*} \log p(Z^*|X) \tag{22}$$

$$= Z_{\mathrm{MAP}} \tag{23}$$

$D_{\mathrm{KL}}\left(q(Z) || p(Z|X)\right)$ is not finite, but its $\arg\max$ and gradient are still well-defined. The same result can be obtained more formally by regarding MAP as the limit of Gaussian mean field inference with diminishing variance $\sigma^2 \to 0$.

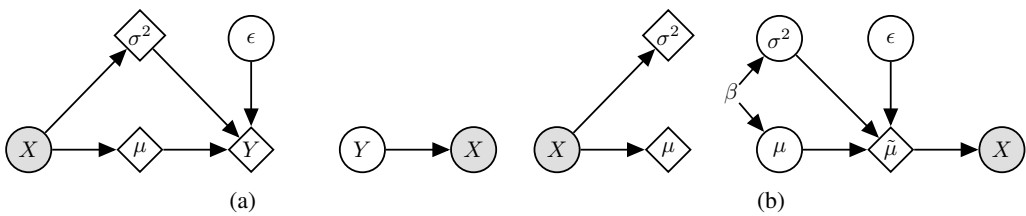

Figure 7: Gaussian mean field inference on a latent model (a) with the ($\beta$-)VAE objective can be reinterpreted as MAP inference on mean and variance latents (b): The output mean and variance of ($\beta$-)VAE's inference model correspond to the latents of the new generative model. This is a per-datapoint-view with data indices $(n)$ and model parameter dependencies omitted.

For improper uniform priors $p(Z) = $ const. we recover Maximum-Likelihood:

$$\arg\max_{Z^*} \log p(Z^*|X) \tag{24}$$

$$= \arg\max_{Z^*} \log p(Z^*) + \log p(X|Z^*) \tag{25}$$

$$= \arg\max_{Z^*} \log p(X|Z^*) \tag{26}$$

$$= Z_{\text{ML}} \tag{27}$$

## C  Latent Capacity in $\beta$-VAE-like Gaussian mean field inference

The purpose of this section is to derive a capacity for flexible-scale Gaussian mean field inference, similar to subsection 3.1. We generalize this to the case when the KL term from the objective is scaled by a factor $\beta > 0$, as done by (Higgins et al., 2017). To illustrate the same capacity bound can be obtained for amortized inference, we derive this capacity for components $Y_i^{(n)}$ of the latents in a $\beta$-VAE, as this is where Gaussian mean field inference is deployed in this case. This is only done fur the purpose of familarity, the same bound would hold for when this inference approach would be applied to model parameters.

We assume Gaussian mean field inference on a latent with a unit Gaussian prior per dimension, as shown in Figure 7a. This analysis includes the standard VAE for $\beta = 1$ as a special case. The $\beta$-VAE objective is (per datapoint, to be summed over all indices $(n)$ which are omitted for readability):

$$F(q, \theta) = \mathbb{E}_{q(Y|X)} \log p(X|Y, \theta) - \beta \sum_i D_{\text{KL}} \left( q(\tilde{\mu}_i) || p(\tilde{\mu}_i) \right) \tag{28}$$

We can write $Y$ in its reparameterized form $Y = \mu^* + \sigma^* \odot \epsilon$, with $\mu^*$ and $\sigma^*$ being the mean and variance yielded deterministically by the inference network and $\epsilon$ being noise sampled from $\mathcal{N}(0, I)$. We can then write the objective as

$$F(\mu^*, \sigma^*, \theta) = \mathbb{E}_{p(\epsilon)} \log p(X|Y = \mu^* + \sigma^* \odot \epsilon, \theta) + \frac{\beta}{2} \sum_i \left( \log \sigma_i^2 - \sigma_i^2 - \mu_i^2 - 1 \right) \tag{29}$$

As in subsection 3.1, we now define a new model featuring NIB that can be trained with a standard variational objective (without need for a factor on the complexity term) so that it results in the same training algorithm (data indices $(n)$ are again omitted). The key idea is to reinterpret the means $\mu$ and variances $\sigma^2$ from the inference distribution as the latents of our new model, as shown in Figure 7b. We keep the deterministic part of the inference network as our new deterministic approximate posterior $q'(\mu, \sigma^2|X)$ with point-mass at $(\mu^*, \sigma^{*2})$ per datapoint and dimension $i$, resembling a MAP approach. We define the priors $\mu \sim \mathcal{N}\left(0, \frac{1}{\beta}I\right)$ and $\forall i : \sigma_i^2 \sim \Gamma\left(\frac{\beta}{2} + 1, \frac{\beta}{2}\right)$ and then define $\tilde{\mu} = \mu + \sigma \odot \epsilon$, where noise $\epsilon$ is sampled from the prior $q'(\epsilon) = p'(\epsilon) = \mathcal{N}(0, I)$. We leave the rest of the model and the inference network unchanged at $p'(X|\tilde{\mu}, \theta) = p(X|Y, \theta)$ and $q'(\mu, \sigma^2|X) = q(\mu, \sigma^2|X)$, and as before, we perform ML learning on the model parameters $\theta$.

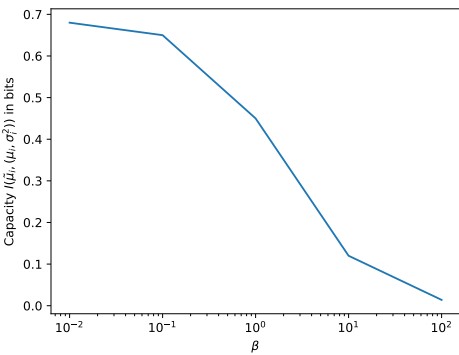

Figure 8: Relationship between $\beta$ and capacity $I(\tilde{\mu}_i, (\mu_i, \sigma_i^2))$ of each latent component in flexible-scale Gaussian mean field inference with complexity term scaled by $\beta > 0$. Values are given in Table 1.

The MAP objective of our newly defined model then is (again per datapoint)

$$F'(q', \theta) = \mathbb{E}_{q'(\mu, \sigma^2 | X) q'(\epsilon)} \log p'(X | \mu, \sigma, \epsilon, \theta) + \sum_i \log p'(\mu_i^*) + \log p'(\sigma_i^{*2}) \tag{30}$$

Equation 29 now implies which is equivalent $F'(\mu^*, \sigma^*, \theta) = F(\mu^*, \sigma^*, \theta) + \text{const.}$, resulting in identical gradients. Because $\epsilon_i$ is sampled from $\mathcal{N}(0, 1)$ in both our method and $\beta$-VAE, the training algorithms are exactly equivalent.

We can now analyze the properties of this reinterpretation of $\beta$-VAE. In particular, we are interested in

$$I(\tilde{\mu}_i, (\mu_i, \sigma_i^2)) = H(\tilde{\mu}_i) - \mathbb{E}_{p(\mu_i, \sigma_i^2)} H(\tilde{\mu}_i | (\mu_i, \sigma_i^2) = (\mu_i, \sigma_i^2)) \tag{31}$$

$$= H(\tilde{\mu}_i) - \mathbb{E}_{p(\sigma_i^2)} \frac{1}{2} \log 2\pi e \sigma_i^2 \tag{32}$$

where

$$H(\tilde{\mu}_i) = -\int_0^\infty \mathrm{d}Y \, p(\tilde{\mu}_i) \log p(\tilde{\mu}_i) \tag{33}$$

$\tilde{\mu}_i | \mu_i, \sigma_i^2 \sim \mathcal{N}(\mu_i, \sigma_i^2)$ with $\mu_i \sim \mathcal{N}\left(0, \frac{1}{\beta}\right)$ implies $\tilde{\mu}_i | \sigma_i^2 \sim \mathcal{N}\left(0, \sigma_i^2 + \frac{1}{\beta}\right)$. Therefore,

$$p(\tilde{\mu}_i) = \int_0^\infty \mathrm{d}\sigma_i^2 \, p(\sigma_i^2) p(\tilde{\mu}_i | \sigma_i^2) \tag{34}$$

$$= \int_0^\infty \mathrm{d}\sigma_i^2 \frac{1}{\Gamma\left(\frac{\beta}{2}\right)} \left(\frac{\beta}{2}\sigma_i^2 e^{-\sigma_i^2}\right)^{\frac{\beta}{2}} \left(2\pi\left(\sigma_i^2 + \frac{1}{\beta}\right)\right)^{-\frac{1}{2}} e^{-\frac{1}{2\left(\sigma_i^2 + \frac{1}{\beta}\right)}\tilde{\mu}_i^2} \tag{35}$$

Figure 8 shows numerical results of Equation 31 for various values of $\beta$. Interestingly, the setting $\beta > 1$ that is suggested by Higgins et al. (2017) for obtaining disentangled representations corresponds to lower latent capacity.

| $\beta$ | $I(Y_i, (\mu_i, \sigma_i^2))$ |
|---------|-------------------------------|
| 0.01    | 0.68 bits                     |
| 0.1     | 0.65 bits                     |
| 1 (VAE) | 0.45 bits                     |
| 10      | 0.12 bits                     |
| 100     | 0.014 bits                    |

Table 1: Numeric results for latent capacities in reinterpreted $\beta$-VAE given by Equations 31, 33 and 35, plotted in Figure 8.

