# OpenReview forum: "Noisy Information Bottlenecks for Generalization"
_ICLR.cc/2019/Conference_

### Official Review · AnonReviewer2 · 2018-11-03
**this paper uses DPI in a wrong way**

**Rating:** 3
**Confidence:** 4

**Review:**

This paper proposes a justification to one observation on VAE: "restricting the family of variational approximations can, in fact, have a positive regularizing effect, leading to better generalization". The explanation given in this work is based on Gaussian mean-field approximation.

I had trouble to understand some parts of this paper, since some of the sentences do not make sense to me. For example

- the sentence under eq. (2)
- the sentence "Bacause the identity of the datapoint can never be learned by ..." What is the identity of a dat point?

It looks like section 2.1 wants to show the connections between eq. (2) and other popularly used inference methods. Somehow, those connections are not clear to me.

Besides some issues in the technical details, the major problem of this paper is that it uses the data processing inequality (DPI) in a **wrong** way.

As in (Cover and Thomas, 2012), which is also cited in this paper, DPI is defined on a Markov chain X -> Y -> Z and we have I(X,Y) >= I(X,Z).

However, based on the definition of \theta and \tilde{\theta} given in the first sentence of section 2.3, the relation between \theta, \tilde{\theta} and D should be: D <- \theta -> \tilde{\theta} (if it is a generative model) or D -> \theta -> \tilde{\theta} (if a discriminative model). Either case, I don't think we can have the inequality in eq. (5).

---

> ### Author Response · Authors · 2018-11-13
> **Response to reviewer 2**
>
> Thank you very much for the constructive review.
>
>
> Summary of our response
> -------------------------------------
>
> We are certain that the data processing inequality is used correctly. As you stated, the DPI implies for any Markov chain X -> Y -> Z that I(X,Y) >= I(X,Z). Unlike suggested in the review, our model is defined in the form \theta -> \tilde{\theta} -> D, as shown in Figure 1a.
>
> Following your feedback, we updated section 2.1 and 2.3 for more clarity.
>
>
> Detailed response
> -------------------------------------
>
> We interleave parts of the review with our detailed response for ease of reading.
>
> > [...] the major problem of this paper is that it uses the data processing inequality (DPI) in a **wrong** way. As in (Cover and Thomas, 2012), which is also cited in this paper, DPI is defined on a Markov chain X -> Y -> Z and we have I(X,Y) >= I(X,Z). However, based on the definition of \theta and \tilde{\theta} given in the first sentence of section 2.3, the relation between \theta, \tilde{\theta} and D should be: D <- \theta -> \tilde{\theta} (if it is a generative model) or D -> \theta -> \tilde{\theta} (if a discriminative model).
>
> Response: We are interested in limiting the mutual information I(\theta, D) between our learned parameters \theta and the dataset D. However, this is hard to calculate for typical deep models. Therefore we introduce a model that forms a Markov chain \theta -> \tilde{\theta} -> D, as shown in Figure 1a. Hereby, \tilde{\theta} is a noisy version of the parameters \theta. Crucially, the data D is defined to be dependent only on the noise-corrupted version \tilde{\theta}. By choosing a convenient noise process and prior for \theta we can easily control I(\tilde{\theta}, \theta). This gives us an upper bound on the mutual information I(D, \theta) between data and parameters, according to the DPI. We updated section 2.3 to reflect this more clearly.
>
> > I had trouble to understand some parts of this paper, since some of the sentences do not make sense to me. For example
> - the sentence under eq. (2)
> - the sentence "Because the identity of the datapoint can never be learned by ..." What is the identity of a data point?
> It looks like section 2.1 wants to show the connections between eq. (2) and other popularly used inference methods. Somehow, those connections are not clear to me.
>
> Response: The aim of section 2.1 is to motivate limiting mutual information for the purpose of generalization. We link generalization problems reported in the literature to the introduced information measure. The information necessary to identify or distinguish between training samples is quantified by the empirical entropy, and we called it the identity of the samples. We updated the section to address all of your feedback.

---

> > ### Comment · AnonReviewer2 · 2018-12-12
> > **about the DPI**
> >
> > Maybe I didn't make my point clear. By saying that this paper uses the DPI in a wrong way, I mean we cannot define an arbitrary chain and claim that it follows the assumption of DPI. The specific way of constructing \theta -> \tilde{\theta} -> D as explained in this paper does not make sense to me.
> >
> > Since the DPI is the foundation of this paper, I don't think this paper is ready to be accepted yet.

---

> > > ### Author Response · Authors · 2018-12-14
> > > **Request for clarification**
> > >
> > > Thank you very much for your response. Unfortunately, it is not clear to us where exactly you see the mistake in our method. We will therefore outline some basic statements about the DPI as well as try to concisely summarize our method and would be very grateful if you could point out a specific statement that you believe to be wrong. We are still convinced that our paper is correct and would like to figure out from which point the misunderstanding arises.
> > >
> > > DPI:
> > > (1) The DPI applies to any Markov chain X->Y->Z for random variables X, Y, Z.
> > > (2) It guarantees that I(X,Y) >= I(X,Z).
> > > (3) If we can construct a probabilistic model that contains such a Markov chain, we obtain a limit on the mutual information as given in (2).
> > >
> > > Noisy Information Bottlenecks:
> > > (4) We consider two use cases with the following probabilistic models:
> > > (4a) Supervised learning having the form \theta->Y<-X. We group (X,Y) into the combined data variable D.
> > > (4b) Unsupervised learning of the form \theta->X. Here the data D is just X. In the paper, we specifically refer to latent variable models with a latent Y, which we can absorb into \theta.
> > > (5) This leads to a common dependence structure \theta->D for both cases, where D represent all data and \theta all learned variables. Our method is applicable to probabilistic models of this general form.
> > > (6) We replace the parameters \theta with a noisy version \tilde{\theta}, such that D is conditionally independent of \theta given \tilde{\theta} and \tilde{\theta} only depends on \theta, giving us a Markov chain \theta->\tilde{\theta}->D.
> > > (7) The DPI applies to this Markov chain, i.e. I(\theta, \tilde{\theta}) >= I (\theta, D).
> > > (8) By choosing the noise distribution and the prior on \theta conveniently, i.e. Gaussian, we can calculate I(\theta, \tilde{\theta}) explicitly, giving us an upper bound on the typically intractable I(\theta, D).
> > > (9) We show that these Noisy Information Bottlenecks are already present in Gaussian mean-field variational inference.

---

> > > > ### Comment · AnonReviewer2 · 2018-12-14
> > > > **About step 5 and 6**
> > > >
> > > > I am glad that you break down the steps, so it is helpful for me to explain my concern, which is mainly from step (5) and (6).
> > > >
> > > > In theory, if you have a chain like X -> Y and you can add anything like Z in the middle and claim the conditional dependence. But my concern is from the way that you actually construct the specific model in the paper.
> > > >
> > > > To my understanding, \theta is the latent variable of the model where D comes from, therefore you have \theta -> D. Based on the description in the paper, \tilde{\theta} also comes from \theta, so we should have \theta -> \tilde{\theta}. If we really want to combine these two components together, we can only get \tilde{\theta} <- \theta -> D.
> > > >
> > > > On the other hand, if you want to show the Markov assumption holds in \theta -> \tilde{\theta} -> D, I would like to see how you define p(D | \tilde{\theta}) precisely without using any information from \theta. By precisely, I mean not the theoretical factorization like p(D, \tilde{\theta}, \theta) = p(D, \tilde{\theta})p(\tilde{\theta}|\theta)p(\theta). The only thing relevant to this question is p'(D | \tilde{\mu}) = p(D | \theta), which I don't think it is correct (regardless the typo).

---

> > > > > ### Author Response · Authors · 2018-12-14
> > > > > **Response**
> > > > >
> > > > > Thank you so much for your prompt clarification.
> > > > >
> > > > > The meaning of \theta depends on the context, for supervised learning it would be the parameters of the model (e.g. a neural network), for a latent variable model in unsupervised learning it would also include the latents (which we have denoted as Y in the paper). The source of confusion here is certainly that \theta -> D is only the model we derive ours from, and we obtain our Markov chain model by **replacing** the original model parameters with a noisy version of \theta, not by appending it. In other words, the model \theta -> D you have in mind is not a part of our model, it is only our starting point from which we motivate our approach.
> > > > >
> > > > > For example in deep supervised learning, we typically have some neural network parameterized by \theta for which we learn point estimates. In contrast, in our model we use a noisy version \tilde{\theta} **in place** of the parameters, and we only learn the mean of this noisy version. Intuitively, the network never sees the exact value of this learned mean, but only noise-corrupted versions of it, which limits how much information we can learn about D. The conditional distribution given by the network stays the same, hence p'(D | \tilde{\mu}) is identical to the original p(D | \theta).
> > > > >
> > > > > To further illustrate this point from another perspective: If you were to generate data from our model, you would first sample a \theta from the prior, sample a noisy version \tilde{\theta} to be used as your network parameters, and finally sample your data from the network (conditioned on some input in the case of supervised learning). We emphasize that once \tilde{\theta} has been sampled conditioned on \theta, the latter is never used again. Hence the Markov property is fulfilled, as the data only depends on \tilde{\theta}.

---

### Official Review · AnonReviewer1 · 2018-11-04
**Interesting ideas, but unclear how to interpret**

**Rating:** 5
**Confidence:** 3

**Review:**

This paper studies "Noisy Information Bottlenecks". The overall idea is that, if the mutual information between learned parameters and the data is limited, then this prevents overfitting. It proposes to create a "bottleneck" to limit the mutual information. Specifically, the bottleneck is created by having the data depend on a noisy version of the parameters, rather than the true parameters and invoking the information processing inequality. The paper gives an example of Gaussian mean field inference. Ultimately, the analysis boils down to looking at a signal-to-noise ratio of the algorithm, which looks very much like regularization.

I think this is a very interesting direction, but the present paper is somewhat unclear. In particular, the example in section 3.1 says that a noisy information bottleneck is introduced, but then says that the modified and unmodified models have "training algorithms that are exactly equivalent." I think this example needs to be clarified. Many of the parameters here are also unclear and not properly defined/introduced. What is the relationship between $\theta$ and $\tilde\theta$ exactly? In this simple model, can we not calculate the mutual information directly (i.e., without the bottleneck)?

The connection between mutual information and generalization has been studied in several contexts [see, e.g., the references in this paper and https://arxiv.org/abs/1511.05219 https://arxiv.org/abs/1705.07809 https://arxiv.org/abs/1712.07196 https://arxiv.org/pdf/1605.02277.pdf https://arxiv.org/abs/1710.05233 https://arxiv.org/pdf/1706.00820.pdf ] and further exploration is desirable. This paper is giving an information-theoretic perspective on existing variational inference methods. Such a perspective is interesting, but needs to be further developed and explained. Specifically, how can mutual information in this context be formally linked to generalization/overfitting? Also, the definition of mutual information used in this paper uses the inferred distribution q (e.g., in eq. 2), which is somewhat unusual. As a result, constraining the model will alter the mutual information and I think the effect of this should be remarked on.

Overall, I think this paper has some interesting ideas, but those need to be fleshed out and clearly explained in a future revision.

---

> ### Author Response · Authors · 2018-11-13
> **Response to reviewer 1**
>
> Thank you very much for the highly constructive review.
>
> > I think this is a very interesting direction, but the present paper is somewhat unclear. In particular, the example in section 3.1 says that a noisy information bottleneck is introduced, but then says that the modified and unmodified models have "training algorithms that are exactly equivalent." I think this example needs to be clarified.
>
> We realized that the naming was very confusing and consequently, we renamed \tilde\theta to \tilde\mu in the noise-injected model. Now,
>  - the original, noise-free model p has the structure \theta -> D (no bottleneck) while
>  - the adapted, noise-injected model p’ has the structure \mu -> \tilde\mu -> D (containing a bottleneck).
> Hereby, \tilde\mu is a noise-corrupted version of the new parameters \mu, and we obtain a limit on the mutual information between \mu and D. We simplified Figure 2 and 8 to make this more clear.
>
> To better characterize Gaussian mean field inference on the original model, we aim to find an inference procedure on p’ so that both algorithms result in exactly the same outcome, e. g. the same calculations are executed when running the corresponding program. We show that there is such an inference procedure on the noisy model, and it has the character of MAP. Note that only if generative and inference model are adapted simultaneously we end up with equivalence. Hereby, \mu (the mean of the Gaussian q) and \theta (the original parameter in p) correspond to \mu (the MAP point-mass of q’) and \tilde\mu (the noise-injected version of \mu in p’).
>
> > Many of the parameters here are also unclear and not properly defined/introduced. What is the relationship between \theta and \tilde\theta exactly?
>
> In this example, \theta and \tilde\theta never appear in the same model (they are part of p and p’, respectively). We realized that this is confusing and have therefore renamed \tilde\theta to \tilde\mu.
>
> > In this simple model, can we not calculate the mutual information directly (i.e., without the bottleneck)?
>
> This is an excellent question. In fact, we believe that trying to construct noise-free deep models with a specific mutual information of data and parameters for the purpose of generalization would be an interesting research direction. Due to nonlinearities in typical deep models, it is at least not obvious how to calculate the mutual information between data and parameters. The main challenge here would certainly be to come up with an effective estimator. Relatedly, one would have to design priors and architecture to achieve a specific mutual information.
>
> > The connection between mutual information and generalization has been studied in several contexts [see, e.g., the references in this paper [...]] and further exploration is desirable. This paper is giving an information-theoretic perspective on existing variational inference methods. Such a perspective is interesting, but needs to be further developed and explained. Specifically, how can mutual information in this context be formally linked to generalization/overfitting?
>
> We updated section 2.2 to relate to the references you mentioned. They explore the link of limiting mutual information and generalization error mostly in theory (and in particular for adaptive analysis). In contrast, we deploy this principle in a practical model structure that is easily applicable to many existing deep and variational learning approaches and provide empirical evidence of the validity of our framework.
>
> >Also, the definition of mutual information used in this paper uses the inferred distribution q (e.g., in eq. 2), which is somewhat unusual. As a result, constraining the model will alter the mutual information and I think the effect of this should be remarked on.
>
> We want to emphasize that we do use the standard definition of mutual information. Therefore, the bottleneck implied by Eq. 5 is purely a property of the generative model and not influenced by the approximate inference distribution q.
> Eq. 2 is only introduced to provide additional motivation for our approach as it allows to characterize overfitting in variational inference. The guarantee derived in section 2.2 ties this quantity back to the mutual information from Eq. 5.

---

### Official Review · AnonReviewer3 · 2018-11-05
**Interesting to read but might lack depth**

**Rating:** 7
**Confidence:** 2

**Review:**

I read the paper and understand it, for the most part. The idea is to interpret some regularization technics as a from of noisy bottleneck, where the mutual information b tween learned parameters and the data is limited through the injection of noise.

While, the paper is a plaisant read, I find difficult to access its importance and the applicability of the ideas presented beyond the analogy with the capacity computation. Perhaps other referee will have a clearer opinion.

I'd be interested to hear if the authors see a connection between their formalism and the one of Reference prior in Bayesian inference (Bernardo et al https://arxiv.org/pdf/0904.0156)

Pro: nicely written, clear interpretation of regularization as a noise injection technics, explicit link with information theoery and Shanon capacity.

Con: not clear to me how strong and wide the implications are, beyond the analogies and the reinterpretation

---

> ### Author Response · Authors · 2018-11-13
> **Response to reviewer 3**
>
> Thank you very much for your encouraging review.
>
> > I read the paper and understand it, for the most part. The idea is to interpret some regularization techniques as a form of noisy bottleneck, where the mutual information between learned parameters and the data is limited through the injection of noise. While the paper is a pleasant read, I find difficult to access its importance and the applicability of the ideas presented beyond the analogy with the capacity computation. Perhaps other referee will have a clearer opinion.
>
> The main contribution of our paper is indeed to establish a connection between variational inference and regularization by observing that Gaussian mean field introduces an upper bound on the mutual information between data and model parameters. Reinterpreting mean field as point estimation in a noisy model allows us to quantify observed regularizing effects. We show links to existing regularization strategies and validate the usefulness for regularization in targeted experiments.
>
> While the focus of our present work lies on establishing links between existing directions of research, we believe that our information-theoretic perspective on regularization opens up plenty of avenues for future work, both in supervised and unsupervised learning.
>
> For example, we are interested in improving extraction of unsupervised representations by controlling the amount of extracted information. In particular, we aim to mitigate latent collapse, a problem reported for example in language generation [1] and autoregressive image generation [2], which is currently mitigated with ad-hoc strategies such as KL annealing. Intuitively, if all information can be stored in the model itself, there is little incentive to use a per-sample latent. This is also known as the information preference problem, as briefly discussed at the end of section 2.1. Therefore, limiting mutual information of the data with the model might offer a robust mitigation strategy. Additionally, we believe that the approach can lead to improved representations through disentanglement, as done by beta-VAE [3]. Our formal connection to beta-VAE derived in Appendix C offers a promising information-theoretic perspective on their empirical results.
>
> More generally, we want to explore non-MAP inference on noise-injected models as this would allow for using highly expressive variational distributions while enjoying the information-theoretic guarantees of simpler approximate distributions, as motivated in section 3.3.
>
> Since these directions are rather orthogonal, we think that sharing our theoretical framework with the community in an independent piece of work is the most effective way of communicating our ideas.
>
> > I'd be interested to hear if the authors see a connection between their formalism and the one of Reference prior in Bayesian inference (Bernardo et al https://arxiv.org/pdf/0904.0156)
>
> Reference priors are opposite to our work in the sense that they maximize the amount of information data provides about the parameters, while we aim to find models to limit it. Also, see [4] for the relation of Fisher information to generalization.
>
> References
> [1] Bowman, S. R., Vilnis, L., Vinyals, O., Dai, A. M., Jozefowicz, R. & Bengio, S. (2015). Generating sentences from a continuous space. arXiv preprint arXiv:1511.06349.
> [2] Gulrajani, I., Kumar, K., Ahmed, F., Taiga, A. A., Visin, F., Vazquez, D. & Courville, A. (2016). Pixelvae: A latent variable model for natural images. arXiv preprint arXiv:1611.05013.
> [3] Burgess, C. P., Higgins, I., Pal, A., Matthey, L., Watters, N., Desjardins, G. & Lerchner, A. (2018). Understanding disentangling in beta-VAE. arXiv preprint arXiv:1804.03599.
> [4] Ly, A., Marsman, M., Verhagen, J., Grasman, R. P. & Wagenmakers, E. J. (2017). A tutorial on Fisher information. Journal of Mathematical Psychology, 80, 40-55, page 30

---

### Meta-Review · Area_Chair1 · 2018-12-15
**Interesting ideas, lacking in clarity.**

**Confidence:** 3
**Recommendation:** Reject

**Metareview:**

The paper proposes a regularization method that introduces an information bottleneck between parameters and predictions.

The reviewers agree that the paper proposes some interesting ideas, but those idea need to be clarified. The paper lacks in clarity. The reviewers also doubt whether the paper is expected to have significant impact in the field.